# A Short Survey on Machine Learning Explainability: An Application to Periocular Recognition

**João Brito** [1,†] **and Hugo Proença** [2,*,†]

1 Department of Computer Science, Faculty of Engineering, Universidade da Beira Interior, 6201-001 Covilhã, Portugal; joao.pedro.brito@ubi.pt
2 IT: Instituto de Telecomunicações, Department of Computer Science, Faculty of Engineering, Universidade da Beira Interior, 6201-001 Covilhã, Portugal
\* Correspondence: hugomcp@di.ubi.pt
† These authors contributed equally to this work.

**Abstract:** Interpretability has made significant strides in recent years, enabling the formerly black-box models to reach new levels of transparency. These kinds of models can be particularly useful to broaden the applicability of machine learning-based systems to domains where—apart from the predictions—appropriate justifications are also required (e.g., forensics and medical image analysis). In this context, techniques that focus on visual explanations are of particular interest here, due to their ability to directly portray the reasons that support a given prediction. Therefore, in this document, we focus on presenting the core principles of interpretability and describing the main methods that deliver visual cues (including one that we designed for periocular recognition in particular). Based on these intuitions, the experiments performed show explanations that attempt to highlight the most important periocular components towards a non-match decision. Then, some particularly challenging scenarios are presented to naturally sustain our conclusions and thoughts regarding future directions.

**Keywords:** periocular recognition; biometrics; explainability; interpretability; machine learning

## 1. Introduction

As our societies become more data-driven, the systems responsible for processing that data have, inevitably, become more complex. Amongst these systems, Machine Learning (ML) models have constituted themselves as innovators in crucial areas, such as medical diagnosis and forensics (law) scenarios. Despite their undeniable prowess, these systems still remain largely latent to the general public and even the developers. In fact, the opaque nature of ML models is one of the reasons that propelled them to the front seat of automatic task solving (i.e., image classification, text generation, and data synthesis) [1]. According to this, a common belief is that interpretability/explainability and performance have an inverse relation: if we want a more transparent system, its performance will likely decrease due to limitations such as the reduction of expressive power for the sake of transparency [2].

The aforementioned argument can be further expanded by analysing the better known ML models and how increases in effectiveness typically lead to more obscure logic. To start, we can look at the most basic family of models, i.e., linear models. In this kind of system, the input features are multiplied by individual weights and the resulting values are added together. It is obvious that features with higher weights are the ones that the model needs the most to reach its prediction. However, because this model can only define hyperlines in the feature space, its applicability is often limited to basic phenomena, yielding significant biases in the generated predictions. To make more useful systems, one could add activation functions, use higher exponents, apply compositions of functions, and so on. Ultimately, we arrive at much more complex (and effective) models, but where their inner rationale

has rapidly lost any intuitive meaning for human understanding. We no longer see a clear path between the input and output, which trades readability for performance.

Consequently, the field of ML interpretability has been emerging to address the concerns above. This paper provides a brief summary of the main techniques used for such purposes, along with a description of some interesting paths for the future.

It is generally accepted that Biometrics constitutes a particularly successful application of ML, with systems deployed worldwide reaching (and even surpassing) human-level performance [3–5]. There is a myriad of traits that can be used to identify individuals, from the classical fingerprint, iris, and face to more recent proposals such as gait or signature dynamics. In this work, we use the periocular region as the main practical case to report/compare the ML explainability features. Using the information in the vicinity (and within) the human eye has been reported as particularly interesting, for performing reliable recognition in low-quality data, acquired under less controlled protocols and environments [6].

The remainder of this paper is organised as follows: To understand the core principles and applicability of interpretability, Section 2 presents the field and its most popular algorithms in more detail, while Section 3 showcases and assesses a method, proposed by this document's authors, which performs the periocular recognition task supported by interpretable explanations. Section 4 discusses the most relevant aspects derived from this work, and Section 5 closes this article, providing some final remarks.

## 2. Machine Learning Interpretability

Fundamentally, the field of ML interpretability comprises techniques and methods that seek to bridge the gap between a system's predictions (i.e., the what) and the reasoning that led to such predictions (i.e., the why). As an introductory example, we could consider the task of training a classifier to optimally discriminate between wolves and huskies [7]. By applying interpretability techniques, we could perhaps notice how the model erroneously uses snow to classify an image as containing a wolf. Without the support of such techniques, it would be decidedly harder to reach the same conclusion. With effect, interpretability brings not only a higher level of trust between the users and a given system, but also a better way of debugging prevailing issues.

Building on the introduction above, it is important to establish some key concepts. In the literature, interpretability techniques are commonly categorised according to the following criteria:

- *Depth*: the degree of complexity that a model is allowed to have. Intrinsic techniques often limit a model's complexity (thus making it more transparent), as opposed to post hoc techniques, which allow complexity and only attempt to explain what the model outputs.
- *Scope*: the range of an interpretability technique. When a technique explains individual predictions, it is operating on a local scale. On the contrary, if a technique allows us to understand a model at once, it is performing a global explanation.
- *Model applicability*: the families of models/architectures that a technique can explain. Model-specific techniques are exclusive to a particular kind of model due to their reliance on specific characteristics of said models. Conversely, model-agnostic methods are generic enough so that they can be paired with virtually any kind of model.

Furthermore, these techniques can be discriminated using different, yet equally legitimate criteria. It is important to evaluate an interpretability method based on its expressive power and complexity. The former is tied to the domain inside which the explanations exist (e.g., natural language, images), while the latter focuses on the computational cost of generating explanations, possibly rendering certain techniques unfeasible. Additionally, the techniques could also be discriminated regarding the properties of the actual explanations they are able to generate. Here, properties such as accuracy (i.e., if the explanations are correct for unseen data), comprehensibility (i.e., the difficulty level of trying to interpret

an explanation), and stability (i.e., the differences between explanations of slightly different samples) stand out.

There are many other properties by which we can assess the usefulness of a given technique [1], but above all else, a technique must be able to make a black-box less opaque and much more user-friendly. The following subsection describes a plethora of visual techniques (i.e., based on images) that can, with varying degrees of success and appeal, achieve such a goal.

### 2.1. Visual Interpretability

In the literature, there are many (and equally valid) forms of interpretability. Techniques such as PDP [8] or ALE [9] present correlations between independent variables and a dependent one via plots, while LIME [10] or SHAP [11] generate images with highlights in certain super-pixels (i.e., groups of neighbouring pixels). For the purposes of this article, the visual explanations convey the highest level of interest. Therefore, the next subsections will cover the better known techniques in the visual domain.

#### 2.1.1. Local Interpretable Model-Agnostic Explanations

From a high-level perspective, LIME focuses on turning on or off certain super-pixels and assessing how a CNN's predictions change based on those perturbations. To do so, we start by taking the sample we wish to explain and create many versions of that image with on or off perturbations. Given this artificial dataset, the black-box model is fed with each of these samples and the respective scores are registered. Then, to understand how the perturbations affect the predictions, the authors proposed the training of an auxiliary (or surrogate) linear model:

$$\text{explanation}(x) = \arg\min_{g \in G} L(f, g, \pi_x) + \Omega(g). \tag{1}$$

Equation (1) shows how an explanation for an instance $x$ is the summation of a loss component $L$ (e.g., MSE) and a complexity component $\Omega$. $L$ measures how the predictions from the surrogate model $g$ and the original model $f$ compare, considering the neighbourhood given by $\pi_x$. $\Omega$ controls the degree of complexity of the auxiliary model (in practice, LIME only optimises the former term).

LIME's disadvantages include the inherent limitations of linear models, which can be incapable of modelling a complex decision boundary, even if it is local and at a smaller scale. Despite the drawbacks, this technique is certainly capable of providing a hint of transparency to otherwise opaque models.

#### 2.1.2. SHapley Additive exPlanations

SHAP has its foundation in Shapley values, which in turn originated from cooperative game theory. Here, features are seen as players inside a potentially collaborative environment, where they can choose to form coalitions (i.e., cooperative parties) to maximise future gains. Due to its flexibility, this technique has served as the basis for many branches. KernelSHAP (referred to as SHAP, for simplicity) is one such branch, and its details will be described further.

Just as LIME, SHAP uses a linear model to approximate the original one. The first step to understanding SHAP is to analyse its loss function:

$$L(f, g, \pi_x) = \sum_{\mathbf{z}' \in Z} [f(h_x(\mathbf{z}')) - g(\mathbf{z}')]^2 \pi_x(\mathbf{z}'). \tag{2}$$

In the equation above, $f$ is the original black-box model and $g$ is the surrogate model, while $\pi_x$ represents the SHAP kernel. Focusing on $g$, the next equation provides the regression form that it naturally possesses:

$$g(\mathbf{z}') = \boldsymbol{\phi}_0 + \sum_{i=1}^{m} \boldsymbol{\phi}_i \mathbf{z}'_i. \tag{3}$$

In addition to what has been stated about $g$, this equation introduces $\mathbf{z}'$, which is a simplified vector of ones and zeros that enable or disable certain features (also known as the coalition vector). Furthermore, $m$ is the maximum coalition size, and $\boldsymbol{\phi}_i$ is the coefficient (i.e., Shapley value) for feature $i$.

Finally, the SHAP kernel demands introduction, given that it is essential in attributing more weight to small or large coalitions, as opposed to coalitions that only aggregate half the features (or close to it). These behaviours are motivated by the notion that we learn more about individual features if we can analyse them in isolation (small coalitions) or if we have almost every feature but one (large coalitions):

$$\pi_x(\mathbf{z}') = \frac{(m-1)}{\binom{m}{|\mathbf{z}'|}|\mathbf{z}'|(m-|\mathbf{z}'|)}. \tag{4}$$

In order to actually train the model $g$, one must start by sampling $k$ coalitions and obtain the original model's predictions when fed these perturbed samples. Then, using Equation (4), each coalition's weights can be obtained, enabling us to use Equation (2) to train the linear model. Once trained, the optimal weights can be seen as the approximate Shapley values for each coalition (note that we can have only one active feature in a coalition or several).

SHAP is a solid technique with comparable results to LIME's, exceeding it in some cases. Additionally, it relies on grounded and proven concepts, such as game theory, Shapley values and LIME's intuitive reasoning. However, SHAP suffers much the same problems as other permutation-based methods: by replacing omitted features with random ones, unlikely data points may be generated, which can lead to unrealistic explanations [7].

2.1.3. Anchors

Proposed in [12] by the same authors that developed LIME, anchors serve similar purposes with different core principles. To start, anchors involve a set of IF-THEN rules so that a prediction is sufficiently anchored by some features (i.e., changes in other features' values do not change the prediction). Amongst the concepts that are introduced with this technique, the notion of coverage is used to specify the amount of instances to which a rule applies (i.e., they are expected to be reusable).

As stated above, the form of an anchor is usually as follows: IF (feature$_1$ == value$_1$ AND/OR feature$_2$ == value$_2$ AND/OR ...) THEN PREDICT target = value$_3$ WITH PRECISION $p$ AND COVERAGE $c$. The readability and user-friendliness are clearly a major concern, given how intuitive the rules are. In practice, for images, the features are the same super-pixels used by the first two techniques, which are then left active or inactive according to their influence on the final prediction.

To determine how precise an anchor is, the authors proposed the following equation:

$$\text{precision}(A) = \mathbb{E}_{D(\mathbf{z}|A)}[\mathbb{1}_{f(x)=f(\mathbf{z})}]. \tag{5}$$

Analysing Equation (5), $\mathbf{z}$ stands for the perturbed neighbours of $x$ to which $A$ is applicable, $D$ is the distribution of perturbed instances, and $\mathbb{1}_{f(x)=f(\mathbf{z})}$ denotes the black-box model's predictions with respect to $x$ and $\mathbf{z}$ (expectedly, the same). In practice, it is intractable to determine adequate anchors using Equation (5). To solve such an issue, the authors proposed the introduction of a new parameter (referred to as $\delta$) such that $0 \leqslant \delta \leqslant 1$, effectively creating a probabilistic definition:

$$P(\text{precision}(A) \geqslant \tau) \geqslant 1 - \delta. \tag{6}$$

Furthermore, the notion of coverage, intuitively explained as the need for rules that are applicable to a large portion of $D$, can also be described with an equation:

$$\text{coverage}(A) = \mathbb{E}_{D_{(\mathbf{z})}}[A(\mathbf{z})]. \tag{7}$$

Maximising coverage is desirable, given that the generated anchors should be reusable on a decently sized portion of the perturbation space:

$$\max_{A \text{ s.t. } P(\text{precision}(A) \geqslant \tau) \geqslant 1-\delta} \text{coverage}(A). \tag{8}$$

From all the equations shown, it becomes clear that this process tries to find anchors with the highest coverage, assuming they satisfy the precision constraint (6). One interesting aspect is that rules with more predicates (i.e., conditions in the IF branch) have a tendency for higher precision. However, such a characteristic is not to be pushed to extreme cases. A rule that has too many predicates may be overly tuned to predict the instance given (*x*) and none other (or a really small amount of similar instances). There is, then, a trade-off between precision and coverage.

Generally speaking, anchors are intuitive, relatively easy to understand, and deliver a performance level close to LIME's, despite not being so popular as the predecessor.

### 2.1.4. Saliency Maps

Described in [13], this technique diverges from the first three counterparts by attempting to highlight certain parts of the input image with brighter or darker shades of white, based on their importance to a model's score (instead of keeping or occluding super-pixels). The authors proposed an introductory example, similar to the following:

$$S_c(\mathbf{I}) = \mathbf{w}_c^T \mathbf{I} + b_c. \tag{9}$$

Equation (9) comes from a linear score model for class *c*, and as is standard, there is a weight vector $\mathbf{w}_c$, which is later transposed, and a bias $b_c$. In this setup, each pixel of image **I** is weighted according to its importance. Translating such a formulation to a CNN, however, becomes more complicated. In spite of this, one can approximate that value by computing the first-order Taylor expansion:

$$S_c(\mathbf{I}) \approx \mathbf{w}^T \mathbf{I} + b. \tag{10}$$

In this case, **w** is the derivative of $S_c$ with respect to a specific image $\mathbf{I}_0$:

$$\mathbf{w} = \left. \frac{\partial S_c}{\partial \mathbf{I}} \right|_{\mathbf{I}_0}. \tag{11}$$

As per Equation (11), we obtain an image-specific class Saliency Map where the magnitude of the derivative indicates the pixels that need to be changed the least to change the final score the most.

Furthermore, and remembering that any image (i.e., $\mathbf{I}_0$) has *m* rows and *n* columns, a class Saliency Map belongs to $R^{m*n}$. The first step is to find **w**, as per Equation (11), through back-propagation. Then, if $\mathbf{I}_0$ is a greyscale image, **w** has exactly one element for each pixel, meaning the Saliency Map could be calculated as $\mathbf{M}_{ij} = |\mathbf{w}_{h(i,j)}|$, where $h(i,j)$ is the index of the element in **w** directly corresponding to $\mathbf{I}_0$'s pixel in row *i* and column *j*. As for RGB images, with multiple depth channels, an index takes the form $h(i,j,c)$, and only the maximum magnitude across all channels is kept: $\mathbf{M}_{ij} = \max_c |\mathbf{w}_{h(i,j,c)}|$.

One advantage of Saliency Maps is that they are not expensive to compute, only requiring a single back-propagation pass, and do not assume the existence of any additional annotations (apart from the labels used when training the original model).

### 2.1.5. Occlusion Maps

Being one of the most straightforward techniques, Occlusion Maps try to make perturbations on certain areas of an input image and register how the model (e.g., a CNN) reacts to those changes.

In practice, this method involves taking a square, of fixed size and colour, and sliding it over the image, iteratively. Then, the perturbed images are fed to the classifier, and its scores are noted. Finally, the scores are translated into a heat map, indicating the portions

of the image where an occlusion had the most impact (i.e., the areas most used by the model to predict the correct class).

In general, Occlusion Maps are able to locate areas of an image that contribute the most to a classifier's decision (albeit not as accurately as other techniques). Still, as an additional benefit, the simple nature of this algorithm means that it is also easy to implement and requires almost no redesign or retraining of the classifier.

## 3. Proof-of-Concept: Deep Adversarial Networks

Taking into consideration the techniques discussed in Section 2.1, we argue that there is still a large amount of uncharted territory in the field of ML interpretability. Therefore, this section goes over a method that attempts to bring interpretability to the task of periocular recognition, specifically, or other similar domains, in general. Considering the four-tier scale suggested in [14], the approach described next could be considered of Level 3, meaning that we leverage domain knowledge (e.g., iris and eyebrow segmentation masks) to make a model's outputs more interpretable. Furthermore, recalling the desirable properties described in Section 2, the present section should clarify the ways in which our explanations are accurate, comprehensible, post hoc, local, and model-agnostic.

### 3.1. Method Overview

The work described in this section was largely based in a previous work of ours [15]. Here, the idea was to produce easily understandable and accurate visual explanations, based on the components that make up the periocular region. As can be seen in Figure 1, our results show red tones for areas that are considered to be really useful for the associated CNN. Conversely, green tones denote areas that are not detrimental.

In order to achieve these results, we started by training a CNN to optimally discriminate between genuine and impostor pairs (therefore performing a verification task). If the query pair was deemed impostor, we entered the explanation stage. In this stage, iris and eyebrow masks were obtained using a trained Mask R-CNN [16] model. Then, given a relatively large dataset of high-quality synthetic genuine pairs (generated with a trained StyleGAN2 [17] generator, on a customised combination of the UBIPr [18] and FFHQ [19] datasets), we found, among these samples, the ones that most closely resembled our query pair, while also having the irises and eyebrows in roughly the same positions.

The key insight of this solution was based in the notion of genuine pair (i.e., two samples from the same subject): as all the synthetic pairs were genuine comparisons, they might look similar to the query pair, but do not generally display the differences that make the query pair an impostor one (such as different iris colours, skin textures, or eyebrow densities). In practice, we obtained a score $s_X$ for each neighbour (the name given to the synthetic samples) using the next equation:

$$s_X = \omega_{\text{masks}} * ||\text{query}_A - \text{neighbour}_X||_2. \tag{12}$$

Once the $k$ best synthetic pairs were found via Equation (12) (and then sorted using that same equation but for images $B$ and $Y$—the second images of the query pair and the neighbour(s)), we obtained the pixel differences between the query pair and such neighbour(s), yielding $k$ intermediate differences (i.e., intermediate images with greyscale tones). Then, these intermediate images were combined into a single one, while giving more weight to the neighbours that looked the closest to the query pair. Finally, interpretable colours were added to the resulting representation, so as to make the explanations more appealing.

### 3.2. State-of-the-Art Comparison

In [15], the framework was evaluated as a traditional recognition method would be (using EER and AUC metrics), culminating in an EER of 0.108 and an AUC equal to 0.813. The experimental setup included the UBIRIS.v2 dataset [20] and the more challenging open-world setting. A fair comparison can be drawn against the results shown in [21] and [22], leading to our belief that our recognition phase consistently outperformed previ-

ous methods. It should be noted that, due to the modular design, our recognition stage could be replaced by another, without compromising the interpretable nature of our results.

As mentioned before, the proposed solution in [15] was not only focused on performing the recognition task well, but, more importantly, on illustrating to the user why certain decisions were made. Regarding the baseline solutions, the following procedure was employed:

- *LIME / SHAP / Saliency Maps*: A DenseNet-121 model was trained to optimally distinguish between genuine and impostor pairs. Upon training, the chosen technique was used in a post hoc setup to explain certain predictions.
- *HL*: Due to this method's natural inclusion of interpretability, it was once again trained to discriminate between the two types of pairs, while at the same time providing heat maps that served as justifications.

Figure 1 shows some illustrative examples of each method's performance:

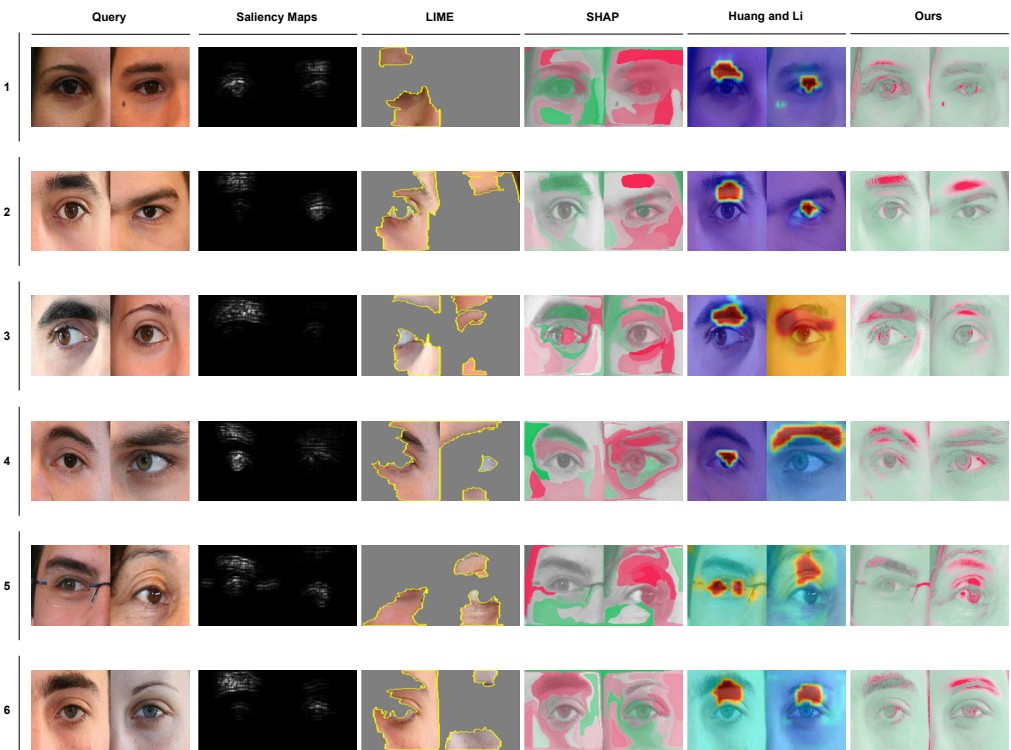

**Figure 1.** Random samples attained by the main methods. Saliency Maps provide greyscale images; LIME keeps or discards super-pixels; SHAP colours them in red (relevant) or green (irrelevant) colours; HL generates heat maps; while [15] replicates SHAP's colour code.

In general, we can make the following comments regarding the results above:

- *Saliency Maps*: The explanations were high level, in the sense that they provided modest highlights for the most obvious differences/components (e.g., eyebrows and eyelids). In our experiments, this approach lacked the ability to emphasise small components that could clearly explain the impostor decision (e.g., person *B*'s skin spots in the first example).
- *LIME*: Despite being one of the most recognisable techniques in the field, LIME only managed to deliver passable results. For instance, irises and spots were never included, despite playing significant roles in Pairs 1 and 4.
- *SHAP*: The colour code was visually appealing, even though the explanations generated by SHAP appeared to be somewhat confusing. There were red and green tones overlapping with each other, and the results could benefit from more consistency

(notice how in Pairs 2 and 3, the eyebrows are coloured in green, in person *A*'s sample, while in person *B*'s, they appeared mostly in red).

- *HL*: Out of all the methods tested, HL seemed to do the best job at being both understandable and consistent. Skin spots, eyebrows, and irises were highlighted where applicable, with the main failures deriving from the lack of highlights in eyebrows (Pairs 2 and 6).
- *Ours*: Taking into account the baseline results, we argue that we were able to match the best of the baseline techniques at worst, or objectively surpassed them at best. Where other techniques failed to highlight spots (Pair 1), eyebrows (Pair 2), or irises (Pair 6), we clearly portrayed them as being used by the accompanying CNN for the impostor class. Another benefit can be seen in the form of fine-grained accuracy, often useful when dealing with smaller components such as skin spots.

## 4. Discussion and Hard Cases

Interpretability is a truly interesting area due to its ability to connect users and AI systems. Throughout this article, a grounded argument was made in favour of adopting existing techniques or developing new ones, to target specific problems with greater transparency. This decision could be enforced by external authorities, the general public, or simply to make the user have a more captivating experience. Considering the existence of the GDPR [23], new software design choices need to be addressed with relative swiftness to ensure compliance with the aforementioned legislation.

Based on what was stated above, we argue that interpretability will likely become a requirement rather than an addition. As shown in Section 3 and [15], interpretability can be incorporated without compromising accuracy or reliability. Even if a newly designed system is not feasible, techniques such as LIME or SHAP can still make black-box models less opaque. It seems to be, then, a question of when and not if, in terms of actual adoption in real-world scenarios.

Furthermore, interpretability could help make safer, fairer, and more efficient algorithms. Debugging a system that is capable of explaining its output is intuitively easier. Making sure that the internal logic is being properly trained and tuned, so as to avoid racial or discriminatory behaviour, is much more important currently given the awareness that people have about such topics.

Based on the benefits above, the next subsection focuses on discussing the improvements that can still be made to elevate the quality of the already useful explanations.

### 4.1. Hard Cases

Our solution [15], LIME, SHAP, Saliency Maps, and Huang and Li's (HL) solution [2] tend to produce less favourable results when faced with certain external factors. Among these variables, there are three particularly interesting cases: background regions, hair and reflections.

When analysing Figure 2, it is important to recall the ways in which the explanations convey information: Saliency Maps are greyscale; LIME's explanations are binary (active super-pixels are relevant, while inactive ones are pointless); SHAP and [15] use red or green tones for important and irrelevant pixels, respectively; HL generates heat maps, in which red areas are of utmost usefulness.

Having established the prelude above, the first pair showed two individuals with different skin and iris colours. However, despite the best efforts of HL and our method to highlight the aforementioned components, the background appeared to be wrongly coloured. Furthermore, in Pair 2, subject *A*'s hair was coloured by SHAP, HL and [15] as being relevant to the impostor class. This behaviour could be considered debatable, given that the hair was in fact different in the sample shown, even if it did not usually belong to the periocular region. Finally, Pair 3 showed a reflection on person *B*'s glasses, which SHAP, HL and our method probably mixed up with some kind of spot or skin mark.

Taking these remarks into consideration, we note how techniques attempting this task need more semantic information to discriminate between traditionally usable components (e.g., iris, eyebrow, sclera, skin, eyelid and spots) and other elements. In the case of the hair

and glasses, for example, the fact that these methods showed them as being important is not necessarily wrong. It is just a matter of either automatically leaving these components out of the explanations or relaxing the problem to allow for more components outside the periocular scope.

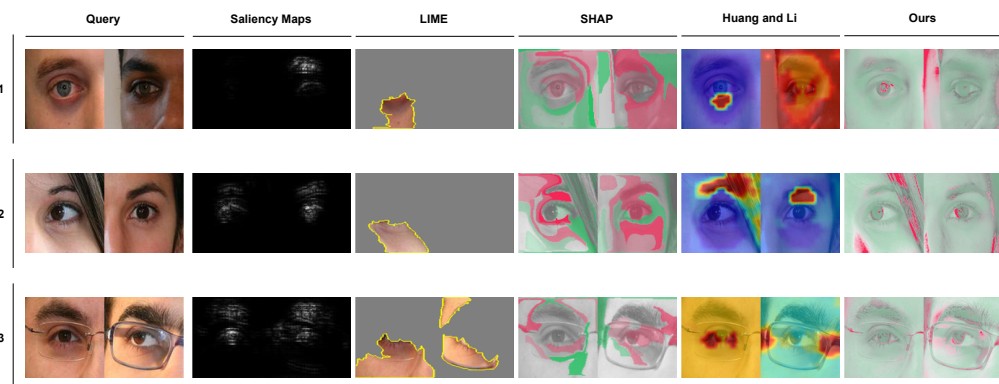

**Figure 2.** Samples in which all the tested methods failed to produce high-quality explanations. Common edge cases include the presence of hair, glasses or background elements.

## 5. Conclusions and Further Work

In this article, an overview of ML interpretability was provided, along with a description of some of the most frequently cited techniques in this topic. Additionally, a method that incorporates interpretability by design was discussed in detail. Overall, a twofold remark can be made: interpretability should be used in as many systems as possible and in the case of visual explanations, techniques such as LIME, SHAP or the method in [15] (for biometric recognition) deliver interpretability to otherwise black-box models. Regardless of how intuitive the results in Section 3.2 may be, some future strides are still needed. In the near future, we expect to apply either generic or model-specific techniques that can show more robustness against occlusions and unwanted components, while still succeeding in the same scenarios as the techniques shown in this document. By addressing the aforementioned limitations, we hope that transparent ML becomes a sustained trend, with obvious benefits for developers and, most importantly, end users.

**Author Contributions:** Conceptualisation, H.P. and J.B.; methodology, J.B.; implementation, J.B.; discussion, H.P.; validation, H.P.; empirical evaluation, J.B.; writing—original draft preparation, J.B.; writing—review and editing, H.P. Both authors read and agreed to the published version of the manuscript.

**Funding:** This research received no external funding.

**Data Availability Statement:** In this article, the publicly available UBIPr (http://iris.di.ubi.pt/ubipr.html, accessed on 30 July 2021), UBIRIS.v2 (http://iris.di.ubi.pt/ubiris2.html, accessed on 30 July 2021), and FFHQ (https://github.com/NVlabs/ffhq-dataset, accessed on 30 July 2021) datasets were used.

**Acknowledgments:** This work was funded by FCT/MEC through national funds and cofunded by the FEDER-PT2020 partnership agreement under the project UIDB/050008/2020. Furthermore, it was supported by operation Centro-01-0145-FEDER-000019-C4-Centro de Competências em Cloud Computing, cofunded by the European Regional Development Fund (ERDF) through the Programa Operacional Regional do Centro (Centro 2020), in the scope of the Sistema de Apoio à Investigação Científica e Tecnológica-Programas Integrados de IC&DT.

**Conflicts of Interest:** The authors declare no conflict of interest. The funders had no role in the design of the study; in the collection, analysis, or interpretation of data; in the writing of the manuscript; nor in the decision to publish the results.

## Abbreviations

The following abbreviations are used in this manuscript:

| | |
|---|---|
| ALE | Accumulated Local Effects |
| AUC | Area Under the Curve |
| AI | Artificial Intelligence |
| CNN | Convolutional Neural Network |
| EER | Equal Error Rates |
| LIME | Local Interpretable Model-agnostic Explanations |
| ML | Machine Learning |
| MSE | Mean-Squared Error |
| PDP | Partial Dependence Plot |
| RGB | Red Green Blue |
| SHAP | SHapley Additive exPlanations |

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
