# Peer review of "A Short Survey on Machine Learning Explainability: An Application to Periocular Recognition"

_electronics, doi:10.3390/electronics10151861_

Round 1
Reviewer 1 Report
This survey paper talks about model interpretability requirements in machine learning and mentions some of the existing methods and techniques. It then shows the examples of interpretation and hard cases in a periocular recognition task. However, it significantly lacks the substances of the technical contributions including ideas, methods and experiments. The example area of the visual models mentioned in the paper is too limited. The survey of the existing work is hence also limited to principle explanations rather than sufficient comparisons of the state of the art work and results in machine-learning model interpretations. It simply describes some of the existing methods in model interpretation and the authors' previous work on a periocular recognition task. The interpretability statement and argument in the conclusion is also not well supported by the discussions and experiments. The overall work written in this paper significantly lacks publishable content to the current research community and is not suitable for Electronics journal.
Author Response
Dear EiC, AEs and reviewers
I are uploading a "pdf" document with all our responses to the comments/suggestions received.
Best regards

Reviewer 2 Report
The topic of this paper is “Towards Visually Explainable Periocular Recognition: A Brief Survey”. The authors would like to introduce the current mainstream for visual interpretability methods. One of the methods was proposed by the author’s group. This method was published in the past (CVPR 2021 workshop). Their method is to propose a framework for visually explainable periocular recognition. The author has also tried to implement those major interpretability technologies, obtained and presented the results. They tried to do some analysis and discussion on the experimental results. At present, the interpretability of artificial intelligence model is still an important research direction, which helps to improve the interoperability between AI and human. Therefore, the content discussed in this article will be a research topic that attracts attention.
However, I hope some more modifications can be made, so that people who want to understand visual interpretability can have a preliminary understanding of the overall technology more quickly. And for different end users, such as business person or model developers, can also get a global view for AI interpretability by reading this article. Here are some of my suggestions:
- Explain more clearly about the technical categories for AI interpretability, so as to give a general view for this problem, not just focus solely on the problem of periocular recognition.
- Describe the advantages and disadvantages of each method evenly. Consider adding a paragraph of explanation or example about the process of applying each method to find visual cues. For example: after summarizing from the reference paper [7] and the author’s practical experiment, the steps that apply LIME in visual interpretation are (1) how to segment the image, (2) how to select the combination of different parts, (3) model prediction results analysis.
3. For the interpretability method (described in section 3) proposed by the author, what are the interpretability characteristics, application value and application scenarios? It’s better to have more description and discussion. For example: among all of the desired characteristics for explainers mentioned in reference paper [7], how many have been satisfied by the proposed method? In addition, what is the major goal of the proposed method? Does it only briefly give explanation for the output of AI model or it can describe more details about the inner operation inside AI model so that it gives more insight for analyzing the internal representation of the model?
4. For the discussion of the advantages and disadvantages of each method, it will be better if the author can insert more experimental results inside the discussion section.
Author Response

(The authors gave the same response as above.)

Reviewer 3 Report
- The paper is very interesting and focused on machine learning explainability, a trendy topic covered in scholarly literature as recently as 2018, for a total of approximately 60 works.
- The abstract section must adjust to the Journal's recommendations, namely:
"Abstract: The abstract should be a total of about 200 words maximum. The abstract should be a single paragraph and should follow the style of structured abstracts,
but without headings: 1) Background: Place the question addressed in a broad context and highlight the purpose of the study; 2) Methods: Describe briefly the main methods or treatments applied. Include any relevant preregistration numbers, and species and strains of any animals used. 3) Results: Summarize the article's main findings; and 4) Conclusion: Indicate the main conclusions or interpretations. The abstract should be an objective representation of the article: it must not contain results which are not presented and substantiated in the main text and should not exaggerate the main conclusions".
In this regard, it is advisable to inform the reader in advance about the results of the periocular recognition experimentation.
- I am not entirely comfortable with the title, as soon as it mentions that it is a "brief survey". I suggest as a more appropriate title "Machine learning explainability: advances in periocular recognition".
- Lines 40-41 state the following: "Biometrics is a particularly successful application of ML, with systems deployed worldwide reaching (and even surpassing) human-level performance". Some references seem appropriate to support this strong assertion.
- There appear to be not many systematic studies on the topic, but perhaps the authors can find additional value for Chapter 2 using the following query:
https://scholar.google.com/scholar?hl=es&as_sdt=1%2C5&as_ylo=2015&as_vis=1&q=intitle%3A%22machine+learning%22+intitle%3A%22explainability%22+intitle%3A%22state%22&btnG=
(Disclaimer: this reviewer is not the author of the aforementioned reference or linked in any way to its origins or stakeholders).
- Lines 217-219 talk about "competitive performance numbers with regard to state-of-the-art methods" from previous work. It is convenient to present these numerical results again in this paper, not only to refer to the previous work.
Final remarks:
1) Consider expanding the Conclusions section to one of Conclusions and future work.
Author Response

(The authors gave the same response as above.)

Round 2
Reviewer 1 Report
Table 1 is completely copied from [15], this is not acceptable.
This paper merely discusses some well-known prior knowledge in the field and some special cases in existing results from the authors' previously published work [15]. The explainability is not proved or justified, it is simply shown as a current process to explain their existing models.
Figure 1 added some new baseline results, but overall there is no new work written in this paper.
Author Response
Dear Associate editor and reviewers
We are enclosing a "pdf" document that describes the changes in the revised version of the manuscript.
Best regards

Reviewer 2 Report
All of the questions have been addressed in this revision.
I think it is OK to be accepted.
Author Response

(The authors gave the same response as above.)
